## RESEARCH ARTICLE

# Past and present goals are represented concurrently during visual search

**Damian Koevoet**[1,2*], **Dirk Van Moorselaar**[1], **Edward Awh**[3], **Stefan Van der Stigchel**[1]

**1** Experimental Psychology, Helmholtz Institute, Utrecht University, Utrecht, The Netherlands, **2** Donders Institute for Brain, Cognition, and Behaviour, Radboud University, Nijmegen, The Netherlands, **3** Department of Psychology, Institute of Mind and Biology, The University of Chicago, Chicago, Illinois, United States of America

* damiankoevoet@gmail.com

## Abstract

Visual selection is often conceptualized as emerging from goal-, stimulus- and history-driven processes within spatial priority maps. Although extensive work detailed the interplay between goal- and stimulus-driven selection, it is largely unknown how goal- and history-driven processes jointly drive selection. While persistent neural firing likely underlies goal-driven selection, it is generally assumed that activity-silent mechanisms effectuate history-driven selection. Due to these different underlying neural mechanisms, simultaneously tracking goal- and history-driven influences neurally has proven difficult. We here employed EEG decoding techniques to simultaneously track and compare goal- and history-driven influences on search. We first established a history-driven signal: Neural decoding closely tracked the target location from the preceding trial. We further demonstrated simultaneous, distinct neural representations of the current and preceding target locations. Strikingly, even when participants attended an upcoming target location before search could commence, prior target locations were reactivated. Our results show that past experiences are reactivated in an inflexible fashion, and do so even when prior targets are completely task-irrelevant. Together, we demonstrate that goal- and history-driven selection are neurally distinct, and reveal that both influences are represented in parallel.

## Introduction

Visual attention has long been conceptualized as emerging from the competition between goal-driven and stimulus-driven processes within spatial priority maps [1–4]. In such maps, attentional priority is weighted across space in a winner-take-all competition that dynamically controls the deployment of spatial attention and gaze. Within this framework, attentional selection results from multiple mechanisms converging within these spatial maps, with top-down goals and bottom-up salience traditionally viewed as the primary determinants of selection priority [3–8].

**Data availability statement:** Scripts to reproduce the reported results are available via the Open Science Framework: https://osf.io/ry683/. The data accompanying the original publications are available here: https://osf.io/nshgd/ and https://osf.io/a9mvb/.

**Funding:** This project has received funding from the European Research Council (ERC; https://erc.europa.eu/homepage) under the European Union's Horizon 2020 research and innovation programme (grant agreement n° 863732 awarded to SVdS). The funders had no role in study design, data collection and analysis, decision to publish, or preparation of the manuscript.

**Competing interests:** The authors have declared that no competing interests exist.

**Abbreviations:** CTFs, channel tuning functions; EEG, electroencephalography; IEM, inverted encoding models; RDMs, representational dissimilarity matrices; RSA, representational similarity analysis.

However, a growing body of evidence has revealed that spatial priority maps are also strongly shaped by selection history—the accumulated influence of past attentional episodes [1,9,10]. Selection history comprises statistically-learned target and distractor regularities [10–14], value-driven attentional capture [15,16], and intertrial priming [17–21]. These history-driven influences operate largely outside conscious awareness, yet they powerfully bias attentional priority toward locations that contained relevant information in the past while suppressing locations that previously contained distractors [14,22–25].

Intertrial priming represents a particularly compelling example of how selection history shapes spatial priority [19,21,26]. This phenomenon entails faster and more accurate responses when target or distractor features or locations are repeated across trials [17–21]. In their seminal study, Maljkovic and Nakayama [21] demonstrated that repeated target locations led to faster responses compared with when target locations changed across trials. Subsequent studies strongly suggest that intertrial priming is driven by changes in spatial priority and not (exclusively) due to more efficient response selection [19,26]. Critically, intertrial priming emerges even when target locations are fully randomized, indicating that these effects cannot be explained by statistically-learned regularities but rather reflect the lingering influence of recent attentional episodes on spatial priority [17–21].

The spatial priority map framework predicts that goal-, stimulus-, and history-based influences should converge on a common priority map. While extensive research has characterized the interaction between goal- and stimulus-driven processes within these maps [3–8,27–29], far less is known about how selection history integrates with these established mechanisms. Although selection history is a strong driver of visual selection, real-world visual search is typically also goal-driven. Thus, to fully comprehend the underlying mechanisms of real-world search, it is crucial to investigate how selection history reconfigures attentional priority when operating alongside top-down goals. This issue has predominantly been addressed with behavioral work, and has led to conflicting results as to how goal- and history-driven selection integrate to shape attentional priority [30–33].

On a neural level, goal- and history-driven selection are thought to recruit distinct mechanisms: While goal-driven selection seems to rely on persistent neural activity [34–37], history-driven selection is generally thought to recruit activity-silent mechanisms by altering short-term synaptic weights [38–43] (but also see [44]). Although history-driven selection is generally not present in ongoing neural activity [39], recent advances show that latent activity-silent states can be accessed through a "pinging" approach [39,45] (also see [46,47]). Pinging entails the presentation of new sensory input to probe the current weights between synapses [45], which multivariate decoders can capitalize on. Nonetheless, ping stimuli are generally task-irrelevant and do not allow for neural tracking of goal-driven selection. While Dolci and colleagues [48] aimed to disentangle the neural correlates of goal- and history-based factors, they could not identify a reliable history-based neural signal, making it difficult to examine how these factors are integrated. Together, even despite recent advances, distinct underlying mechanisms made it particularly difficult to neurally track goal- and

history-driven influences simultaneously, leaving fundamental questions about their putative interplay within spatial priority maps unresolved.

We here aimed to isolate distinct neural signals tracking goal- and history-driven selection to directly examine their time-resolved operations within spatial priority maps. Instead of relying on task-irrelevant ping probes, we here reasoned that search arrays should carry enough stimulus energy to allow for the neural tracking of history-driven selection [23]. This comes with the benefit of allowing for additional tracking of goal-driven selection. To this end, we applied decoding techniques to electroencephalography (EEG) data from two prior visual search studies [34,35], using intertrial priming as an exemplar of history-driven selection.

Prefacing our results, we first found substantial intertrial priming effects on search: responses were faster and more accurate when target locations were closer to one another in successive trials—even though target locations were chosen randomly. We then established a history-based neural signal by demonstrating that previous target locations were consistently reactivated despite being task-irrelevant. To further examine the interplay between goal- and history-driven influences on spatial priority maps, we turned to a cued visual search task. We found that even when top-down spatial attention was deployed to a future target location, previous target locations were reactivated. Moreover, representational similarity analysis (RSA) revealed concurrent, distinct representations of previous and current target locations. Together, our findings provide strong neural evidence that goal- and history-driven influences operate through separable mechanisms within priority maps, with both simultaneously represented in the brain.

## Results

### Intertrial priming affects search in a spatially graded fashion

In Experiment 1, 23 participants searched for a vertical or horizontal bar among distractors while scalp EEG was recorded (Fig 1A). In alternating blocks, the difficulty of the search task was manipulated by altering the properties of the distractor stimuli [34,49].

We first analyzed behavioral data to ascertain whether the previous target location affected current search (Fig 1B and 1C). To this end, we split the data based on the distance between the current and the previous target location. Response times were analyzed using a two-way repeated-measures ANOVA (2 task × 5 intertrial target distance) (Fig 1B). As expected, responses were faster when the previous target location was close to the current target location ($F(2.11, 46.33) = 53.97$, $p < .001$, $\eta_p^2 = .710$). We also observed slower responses in the hard compared with the easy search condition (as reported previously; $F(1, 22) = 155.26$, $p < .001$, $\eta_p^2 = .876$). The interaction effect was also significant ($F(4, 88) = 33.98$, $p < .001$, $\eta_p^2 = .607$). To explore this interaction, we fit a linear regression for each condition per participant. We found that when the prior and current target were closer to one another, responses were faster in both conditions (one-sample t-tests of slopes against 0; easy: $t(22) = 5.41$, $p < .001$, $d = 1.13$, hard: $t(22) = 8.65$, $p < .001$, $d = 1.80$). Moreover, this spatial gradient was stronger for the hard than the easy condition ($t(22) = 7.57$, $p < .001$, $d = 1.58$).

We observed a comparable pattern of results when analyzing accuracy (Fig 1C), effectively ruling out speed-accuracy trade-offs. Accuracies were higher when target locations were closer to the previous target location ($F(2.33, 51.32) = 7.10$, $p = .001$, $\eta_p^2 = .244$). As reported previously, accuracy was higher in the easy than in the hard search task ($F(1, 22) = 20.93$, $p < .001$, $\eta_p^2 = .488$). The interaction effect between task and intertrial target distance was also significant ($F(2.52, 55.35) = 6.24$, $p = .002$, $\eta_p^2 = .226$). We found a significant spatial gradient in the hard ($t(22) = 2.93$, $p = .008$, $d = .61$) but not in the easy condition ($t(22) = 1.27$, $p = .216$, $d = .26$), and accuracy slopes were significantly steeper in hard compared with easy blocks ($t(22) = 2.51$, $p = .020$, $d = .52$). The lack of a spatiall gradient in accuracy in the easy condition could, however, be attributed to a ceiling effect (i.e., the task was too easy overall). Together, the behavioral results revealed strong intertrial priming effects on search.

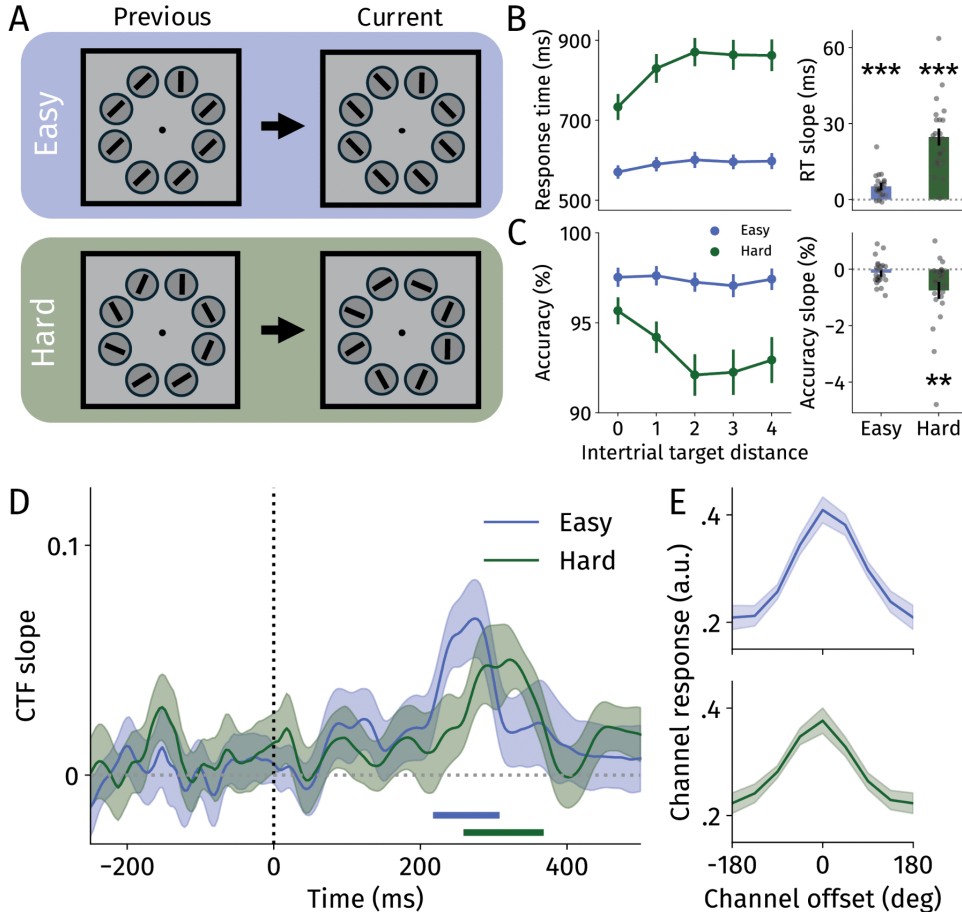

**Fig 1. Task and results from Experiment 1. A** Visual search task displays. Participants searched for a vertical ("z" key) or horizontal ("/" key) target among distractors in an easy and hard condition. Dimensions do not match those in the experiment, but have been changed here for clarity. **B** Response times separately for the easy (blue) and hard (green) search conditions on the left. Response times are plotted as a function of distance between the current and previous target location. On the right: slopes (i.e., beta) calculated using linear regressions per participant to assess the spatial gradient of intertrial priming. Black dots represent individual participants. **C** Same as for B, but for accuracy data. See S1 Data for individual data points. **D** CTF slopes locked to array onset over time for the easy (blue) and hard (green) conditions. Horizontal blue and green lines represent significant clusters (i.e., CTF slopes above 0; $p < .05$) for the easy and hard conditions, respectively. **E** Reconstructed CTFs of the easy and hard conditions averaged across their respective significant time windows. A channel offset of 0 corresponds to the previous target location. All error bars and bands reflect standard errors of the mean. RT, response time. **$p < .01$, ***$p < .001$.

## Neural reactivation of previous targets

We then asked whether neural activity tracked the target location from the preceding trial to establish a history-driven neural signal (Fig 1D and 1E). To this end, we employed inverted encoding models (IEM) [50,51] which have previously been used to precisely track spatial attention across time [34–37,52]. We here instead trained the models to decode the target location from the preceding trial using voltage data.

We were able to precisely track the previous target location ($n − 1$) shortly after array onset in the easy condition (242–342 ms; $p = .008$). Similarly, we could track the previous target location in the hard condition (286–406 ms; $p = .007$). Examining reconstructed CTFs averaged across the significant clusters (Fig 1E), reveals clear spatial selectivity for the previous target location in both conditions. Since we discarded trials wherein the current target location matched the previous target location for this analysis, these results cannot be attributed to decoding of the current target location (also

see S1 Fig). There were no significant differences between search conditions ($ps > .39$). Together, we found that previous target locations were transiently reactivated in two types of search.

One alternative possibility is that previous target locations were electrically active prior to search array onset, but appear as a transient reactivation due to the application of a baseline correction. We ruled out this possibility by calculating changes in voltage from sample to sample (i.e., derivative), which does not require baseline correction, and reconducted the IEM procedure on this data (S2 Fig). We found that voltage changes after stimulus onset revealed the previous target location. This is strongly indicative of a reactivation, and is consistent with an activity-silent account of history-driven selection.

In a series of control analyses, we ruled out that eye movements drove the spatial selectivity in the reconstructed CTFs throughout the paper (S3–S5 Figs). These analyses showed that reactivation of previous target locations was not driven by eye movements but by neural activity.

As behavioral intertrial priming effects typically extend beyond the previous trial [19,21], we also tested whether the target location from two trials ago ($n − 2$) was reactivated. We found no reliable spatial selectivity to the target location from two trials ago across search conditions (S6 Fig). This result could be due to the fact that only more recent target locations are strongly reactivated or because EEG is not sensitive enough to pick up on previous experiences across longer intervals [43].

## Neural reactivation despite covert spatial selection

The above analyses demonstrated that prior target locations are reactivated (Fig 1), even though these are unpredictive of, and therefore uninformative for the upcoming search task. We pushed this further and tested whether prior target locations reactivated when participants knew with certainty where the upcoming target location would be presented.

To address this, we analyzed data from a cued visual selection task [35]. In Experiment 2, 42 participants identified whether the gap was on the left or right of the target diamond (Fig 2A). Prior to search onset, participants either received an informative cue indicating the target location (100% valid) or a neutral cue. Thus, participants were able to deploy covert spatial attention toward a target location prior to search onset in the informative but not in the neutral condition.

Responses were faster and more accurate in covert than neutral blocks, indicating that attention was shifted to the target location prior to search onset (speed: $F(1, 41) = 186.30$, $p < .001$, $\eta_p^2 = .820$; accuracy: $F(1, 41) = 56.43$, $p < .001$, $\eta_p^2 = .579$; as reported previously). In line with Experiment 1, we found faster ($t(41) = 9.07$, $p < .001$, $d = 1.40$) and more accurate responses ($t(41) = 3.19$, $p = .003$, $d = .49$) when previous and current targets were closer to one another in neutral blocks. By contrast, we observed only a modest effect of intertrial priming on accuracy ($t(41) = 2.21$, $p = .033$, $d = .34$) and no significant effect on response speed in covert blocks ($t(41) = 1.47$, $p = .149$, $d = .22$). While accuracy slopes did not differ significantly between conditions ($t(41) = 1.14$, $p = .263$, $d = .18$), response time slopes were significantly flatter in covert than in neutral blocks ($t(41) = 4.09$, $p < .001$, $d = .63$). These results revealed that goal-driven covert spatial selection attenuated intertrial priming effects in search.

Turning to the neural data, we aimed to track the previous target location using IEMs as in Experiment 1. In neutral blocks, we observed a brief reactivation of the previous target location upon cue onset (172–426 ms; $p = .014$). Upon search array onset, the previous target location was again reactivated (876–1,200 ms; $p < .001$) - in line with Experiment 1. In covert blocks, we found that—despite goal-driven selection—the previous target location was reactivated upon cue onset, and continued to be represented throughout the trial (114–1,200 ms; $p < .0001$). For a brief period following array onset, the previous target location was more strongly represented in the covert compared with the neutral condition (804–932 ms; $p = .047$). Our results reveal that even when participants deployed covert spatial attention (using a 100% valid cue), prior target locations were neurally reactivated. This demonstrates that neural reactivations of past experiences are inflexible and occur even when wholly task-irrelevant, or could even be detrimental to selecting task-relevant information.

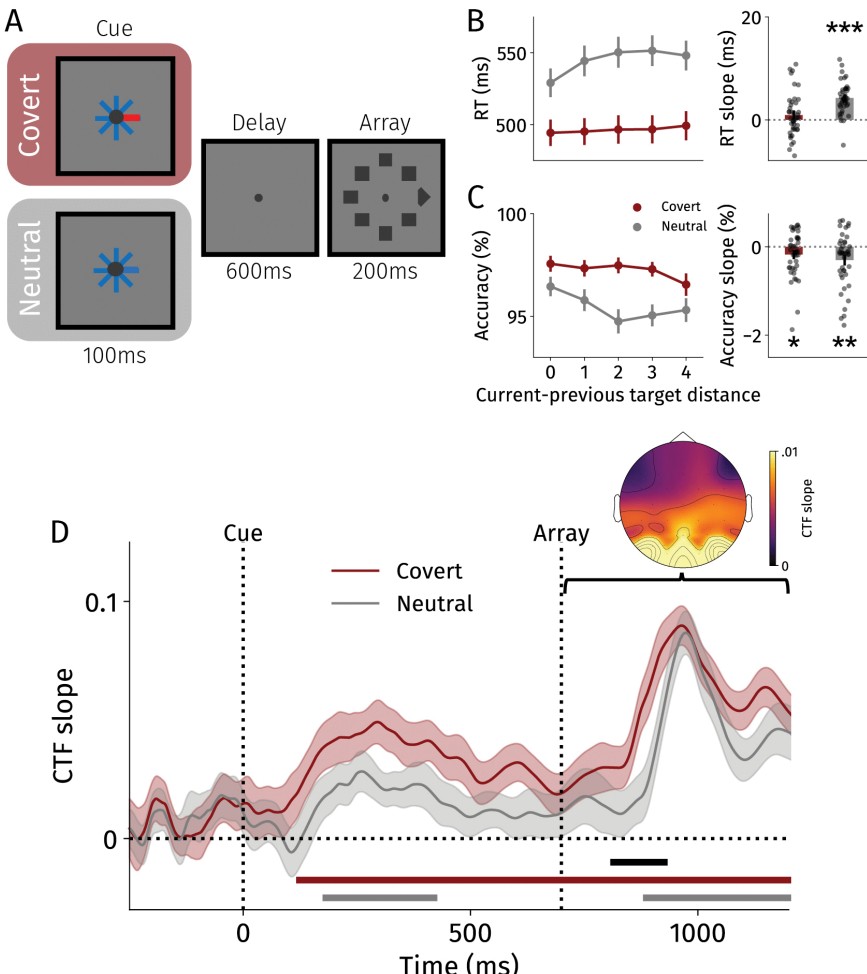

**Fig 2. Task and results from Experiment 2. A** Cued visual selection task. Participants searched for the gap in the target diamond in covert and neutral conditions. Dimensions do not match those in the experiment but have been changed here for clarity. **B** Response times separately for the covert (red) and neutral (gray) conditions on the left. Response times are plotted as a function of distance between the current and previous target location. On the right: slopes (i.e., beta) calculated using linear regressions per participant to assess the spatial gradient of intertial priming. Black dots represent individual participants. **C** Same as for B, but for accuracy data. See S2 Data for individual data points. **D** CTF slopes (i.e., spatial selectivity) from cue onset over time for the covert (red) and neutral (gray) conditions. Topography reflects the outcome of the searchlight analysis averaged across covert and neutral conditions, where higher values indicate that an electrode elicited a steeper CTF slope. Horizontal red and gray lines represent significant clusters (i.e., CTF slopes above 0; $p < .05$) for covert and neutral conditions, respectively. Horizontal black line represents a significant cluster-corrected difference between conditions ($p < .05$). All error bars and bands reflect standard errors of the mean. RT, response time. *$p < .05$, **$p < .01$, ***$p < .001$.

To further characterize the reactivation pattern, we conducted a searchlight analysis [53] to visualize the contribution of each electrode across the scalp to our IEM results [37]. To this end, we iteratively selected groups of four or five neighboring electrodes across the scalp and reconducted the IEM procedure. We calculated an electrode's spatial selectivity to the previous target location by averaging across all electrode groups that an electrode was a part of. This way, we were able to visualize spatial selectivity for each electrode while collapsing across all eight target locations. To facilitate visualization, we averaged CTF slopes upon array onset (700−1,250 ms after cue onset), and as we found highly comparable results for the covert and neutral conditions, we averaged topographies across conditions. Closely resembling topographies tracking current target locations [37,54], we found that especially posterior channels tracked the location of the previous target location.

## Simultaneous representations of previous and current targets

Until now, we exclusively focused on neurally tracking the location of previous targets, and disregarded the location of current targets. To investigate the competition between previous and current target locations, we employed time-resolved representational similarity analysis (RSA) [55] which allowed us to isolate neural activity that tracked previous and current targets over time (see Materials and methods).

We constructed theoretical representational dissimilarity matrices (RDMs; Fig 3) for previous and current target locations that predicted neural activity to be more similar when locations are closer to each other. For example, while neural activity should be relatively similar when representing right and top-right locations, neural activity should be relatively dissimilar when representing right and left locations [56–58]. Following prior work [52,59], we calculated cross-validated Mahalanobis distance (i.e., a measure of multivariate distance) for each combination of previous and current target locations to quantify neural similarity [60]. To isolate the effects of previous and current target locations on neural similarity, we calculated semi-partial rank correlations (as we did not assume linear relationships) [52,61]. We then assessed whether and when previous and current targets were neurally represented by testing whether semi-partial rank correlations were reliably above 0 across participants. This allowed us to directly test whether goal- and history-driven influences explained distinct variance in ongoing EEG activity.

In Experiment 1, we found that RSA tracked previous and current target locations upon search onset (Fig 3B). Specifically, in the easy search condition we found that previous (90–430 ms; $p < .001$) and current (210–500 ms; $p < .0001$) target locations were simultaneously represented shortly after array onset. We similarly found concurrent representations of the previous (290–390 ms; $p = .015$) and current (130–500 ms; $p < .0001$) target locations when search was more difficult in the hard condition. Across both conditions, we observed that current targets were represented more strongly than previous targets. Notably, semi-partial correlations were considerably lower for the target location in the hard compared with the easy condition. We speculate that this can be attributed to the more variable response times in this condition, which reduces the temporal alignment of neural signals and consequently impaired decoding accuracy (in line with [34]). In particular, if the latency of finding the target is more variable (e.g., because of selecting a distractor on some trials), neural representations of the target location will also be more variable across time, which may account for the less pronounced target representation in the hard condition.

In the neutral condition (Fig 3C), the target location was represented from array onset (790–1,200 ms; $p < .0001$). Although numerically the data fit the pattern obtained from the IEM analyses, we found that the previous target location was not significantly reactivated after cue (170–250 ms; $p = .090$) or array onset (1,050–1,130 ms; $p = .103$). As a less stringent test, we averaged semi-partial correlations from array onset (700–1,200 ms post-cue) and conducted a one-sided one-sample $t$ test. This showed a modest but significant reactivation of the previous target upon search onset ($t(41) = 2.16$, $p = .018$, $d = .33$).

In the covert condition (Fig 3C), we observed that the current target location was continually represented neurally from cue onset throughout the trial (70–1,200 ms; $p < .0001$). Aligned with our IEM analysis, we found that the previous target location was also represented after cue onset (70–610 ms; $p < .001$), and was reactivated upon search array onset (690–1,200 ms; $p < .001$). This provides strong evidence that even when participants deploy covert spatial attention to a target location, the previous target location is also reactivated.

One potential limitation of the decoding of the current target location in the covert condition is that the cue stimulus itself carried spatial information. However, we believe that the specific properties of the cue cannot account for the sustained decoding of the current target location for multiple reasons. First, participants responded faster and more accurate in covert compared with neutral trials, indicating that attention was deployed prior to search array onset. Second, cue stimuli were small in size (0.125°x 0.08°) making it unlikely that their weak sensory energy allowed for sustained decoding throughout the delay period. Third, recent work using the same cue stimuli demonstrated differences in spatial selectivity between preparatory overt and covert shifts of attention [37], indicating that the neural decoders are picking up

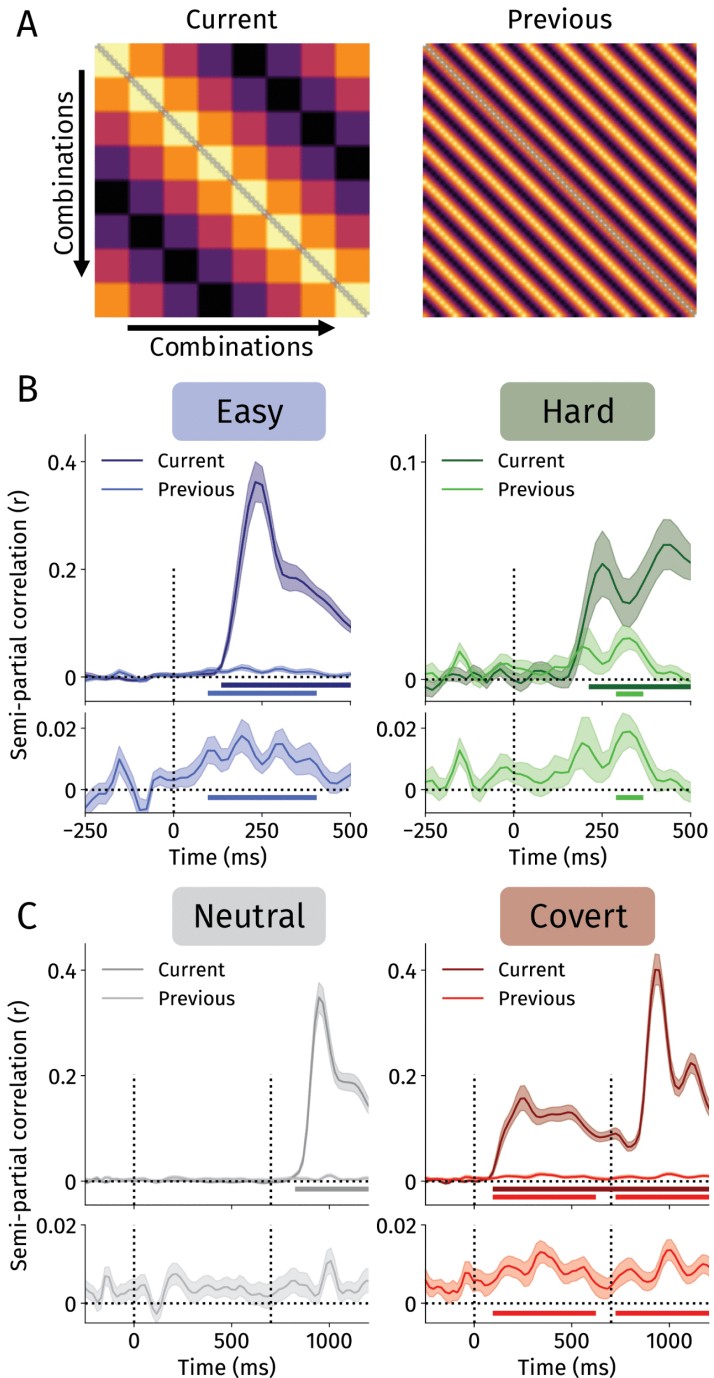

**Fig 3. Representational similarity analyses. A** Theoretical representational dissimilarity matrices. Matrices are built from combinations of the previous and current target locations (in total 8 previous x 8 current targets = 64 combinations). By using combinations of previous and current targets, we were able to construct independent theoretical dissimilarity matrices for previous and current target locations. Each cell corresponds to the similarity of two combinations in terms of the current or previous target. Specifically, similarity in these matrices was the distance between targets (ranging between 0 and 4), where smaller values indicate similar locations and vice versa. Bright locations indicate a higher similarity across locations (and vice versa). **B** RSA results for Experiment 1. On the left: Semi-partial correlations in the easy condition (blue) for the current and previous target over time. Below, the previous target semi-partial correlations are replotted to ease inference (it is the same data as above). On the right: The same, but now for the hard condition (green). **C** Same as for B, but now for Experiment 2. Neutral and covert data are plotted in gray and red, respectively. Horizontal lines represent significant clusters (i.e., semi-partial correlations above 0; $p < .05$). All error bands reflect standard errors of the mean.

on attentional processes, likely alongside transient spatially selective sensory input provided by the cue stimulus. These arguments support the conclusion that decoding of the current target location reflects the deployment of covert spatial attention.

The RSA results are largely in line with the IEM analyses reported above, and additionally demonstrate that both goal- and history-driven influences can be tracked independently and concurrently. Indeed, RSA allowed us to track and compare previous and current search targets on equal footing, despite their distinct underlying neural mechanisms. Moreover, RSA showed that previous and current target locations predicted distinct variance in neural activity, providing strong evidence that these are independent factors influencing visual selection. We consistently found that current target locations were more strongly represented than previous target locations, but were activated simultaneously. Together, our results demonstrate that goal- and history-driven influences are driven by distinct neural signals that are simultaneously represented in the brain.

## Discussion

We here provide strong evidence that goal- and history-driven influences explain distinct variance in neural activity, while both have parallel influences on behavior. We applied multivariate neural analyses to multiple visual search datasets which consisted of more than 100,000 trials in total. We consistently observed transient, spatially selective neural reactivation of the previous target location. Such reactivations were inflexible: they even occurred when participants deployed covert spatial attention toward the upcoming target location preceding search onset. Further analyses allowed us to track and compare distinct neural signals carrying goal- and history-driven information on equal footing. This showed that the current target is more strongly represented than the previous target, yet both were represented concurrently.

Activity-silent accounts of working memory [38,47,62,63] state that dormant representations of past experiences are reactivated only when they become task-relevant. As such, reactivation occurs prior to a memory report [47] or when a specific item may still be probed at the end of the trial [46]. Aligned with this, we find support for reactivations of past experiences across both experiments. Yet, our data demonstrate that such reactivations of past experiences occur even when completely task-irrelevant. Indeed, we found that although target locations were random, previous target locations were neurally reactivated, which is in line with previous work [23,39–44]. We extended this further and found that previous target locations were reactivated even when covert spatial attention was deployed to the upcoming target prior to search onset (Figs 2D and 3C). This indicates that not only activity-silent representations that are useful for current behavior (e.g., working memory item) but also task-irrelevant past representations are reactivated [43]. Complementing this, recent work demonstrated that statistically-learned regularities guide attention rather inflexibly as regularities transfer between tasks and operate in retinotopic (as opposed to spatiotopic) space [64,65]. Activity-silent mechanisms underlying memory and history-based selection may thus act less flexibly and adaptively than previously thought.

Our data provide strong evidence that participants had concurrent, distinct neural representations of the current and previous target locations. Many theories of visual attention pose that independent underlying layers together shape an integrated priority map [1–4]. Goal-, stimulus-, and history-based factors are thought to paint on independent layers that together make up the attentional priority map that decides where attention is deployed in a winner-take-all competition. Using RSA, we were able to simultaneously track the goal- and history-based layers of the priority map across time. We showed that current targets were more strongly represented than previous targets. In addition, we were also able to pinpoint when these layers became active and demonstrated that both layers were represented in the brain simultaneously. While previous targets were only briefly represented, current targets remained active throughout the trial. This demonstrates that previous and current targets both exert their influence on selection, until the target is identified, selected, and ultimately acted upon. In parallel, we observed intertrial priming effects on search across experiments, suggesting that we here uncovered the neural mechanisms reflecting the time-resolved interplay of goal- and history-driven selection.

Moreover, although the behavioral results from Experiment 2 indicated that goal-driven selection using the spatial cue attenuated history-driven signals, the neural data revealed that previous target locations were still continually represented. Thus, our data demonstrate that even when selection history does not affect search strongly, it is still operating on a neural level. Together, our results provide strong neural evidence that top-down control and selection history exert distinct influences on visual selection [1,2]. Future research may adopt our approach to study how the confluence of goal-, stimulus-, and history-driven [1–3,5,6] (and effort-driven; [66,67]) factors guide visual selection concurrently.

While our findings possibly demonstrate that task-relevant search arrays can serve as "ping" stimuli to reveal latent activity-silent states [23,39,45], the current results also point to an alternative explanation: history-driven signals may emerge through operational engagement of the priority map rather than through sensory perturbation per se. Instead of requiring external sensory perturbation to access latent activity-silent states, history-driven representations may be auto-matically co-activated whenever the priority map is updated or recruited for search initiation. Several aspects of our data support this "operate-to-reactivate" account. Most notably, informative cues reliably evoked spatially selective neural activ-ity to prior target locations (Fig 2), despite their minimal sensory energy—arguably insufficient to act as ping stimuli. Such reactivations may have emerged because the priority map was engaged during covert trials upon cue onset, which also accounts for the (somewhat) stronger reactivations compared with neutral trials. Moreover, previous target reactivation onset occurred earlier in covert versus neutral conditions, where priority map engagement likely occurs sooner following array presentation. Similarly, this accounts for the differential reactivation latencies between easy and hard conditions (Fig 1C): if participants engage the priority map (i.e., by updating the goal-driven layer) earlier in easy trials where targets "pop out," history-driven reactivations should emerge correspondingly earlier compared with hard conditions.

The operate-to-reactivate framework suggests that the priority map has an integrated representational structure [4, 68, 69], where engaging one layer of the map reveals information embedded in other layers. We here provide the first tentative evidence for this by showing that engagement of the goal-driven layer reactivated latent history-driven informa-tion. If the priority map indeed has an integrated representational structure, activation of the stimulus-driven layer should also reveal history-driven information, which accounts for previous results obtained with pinging paradigms [23,39]. Notably, however, recent work demonstrated that goal-driven attention can be engaged during the presentation of pinging displays [70], which could have inadvertently activated the integrated priority map rather than merely probing latent states through sensory perturbation (see [23]). While further research is needed to establish the boundary conditions for history-driven reactivations, our findings demonstrate that goal- and history-driven influences are represented simultane-ously and distinctly in the brain, regardless of the precise underlying mechanism.

We here investigated history-driven selection using intertrial priming as an exemplar. One advantage of this approach is that intertrial priming generally exerts large behavioral effects (Figs 1 and 2), and that it occurs in virtually all search experiments. Nevertheless, it remains unclear whether other facets of selection history, such as statistically learned target/ distractor regularities, follow similar time courses as those observed here. Duncan, van Moorselaar, and Theeuwes [39] could decode a statistically learned and previous target location prior to search onset with similar time courses using a pinging approach [45] (also see [23]). Although this argues that intertrial priming may function as a good exemplar for selection history as a whole, work incorporating different aspects of selection history is necessary to generalize our find-ings further.

We here applied multivariate neural decoding to visual search data from multiple experiments to concurrently track current and previous target locations. Our analyses consistently demonstrated that activity-silent representations of past target locations are neurally reactivated even though these locations are now wholly task-irrelevant. Our results provide strong neural evidence that goal- and history-based factors have separate underlying layers making up the attentional priority map, and that both of these layers were simultaneously represented in the brain. Altogether, our results provide critical evidence that top-down control and selection history are independently yet simultaneously repre-sented in the brain.

## Materials and methods

Scripts to reproduce the reported results are available via the Open Science Framework: https://osf.io/ry683/. The data accompanying the original publications [34,35] are available here: https://osf.io/nshgd/ and https://osf.io/a9mvb/.

### Ethics statement

Procedures from Experiments 1 and 2 were approved by the institutional review boards at the University of Oregon and the University of Chicago, respectively. All participants provided written informed consent prior to taking part in the experiments.

### Participants

Across all our analyses, we included data from the same participants as those included in the original publications [34,35]. Data from 23 (Exp. 1) and 42 (Exp. 2) participants with normal or corrected-to-normal vision were analyzed. We expected to have sufficient statistical power due to the high number of trials per participant, and also because the current sample size exceeds [43] or was comparable to existing work [39] using EEG decoding to investigate history-based effects.

### Procedure

As we here report novel analyses of previously published data [34,35], we briefly describe the experimental procedures and refer the reader to the original publications for precise details. Note that we here report Experiment 2 from Foster and colleagues [34] as Experiment 1. In all analyzed data [34,35], target locations were randomized, so repeating target locations occurred solely by chance. Thus, participants had no incentive to prioritize previous target locations because each location had an equal target probability.

**Experiment 1.** Participants performed a visual search task wherein they searched for a vertical or horizontal bar among distractors (Fig 1). At the start of each trial, eight dark gray background circles were presented, and the target and distractor lines were presented on top of these circles. Participants were instructed to indicate whether the target was vertical ("z" key) or horizontal ("/" key) as quickly and accurately as possible. Feedback regarding accuracy and response speed was provided at the end of each block. Search arrays were presented for 2 s, and intertrial intervals ranged between 1.8 and 2.3 s.

The visual search task consisted of easy and hard conditions, alternating across blocks. Distractors in the easy condition always had the same orientation on a given trial, and distractor orientations were always 45° from possible target orientations. The hard condition was more difficult because distractors could vary in their orientation and distractor orientations were closer to possible target orientations (22.5°) [49].

The experiment consisted of a total of 30 blocks, each containing 64 trials. The starting condition was counterbalanced across participants.

**Experiment 2.** Participants performed a cued visual selection task wherein they searched for a target diamond that had a gap on the left ("z" key) or on the right ("/" key) (Fig 2A). In alternating blocks, participants either received a spatial cue indicating the upcoming target location, or they received a noninformative cue. Thus, in half of the trials participants could preemptively shift covert spatial attention toward the target location prior to search onset.

Each trial started with a fixation period ranging from 1.5 to 1.9 s (Exp. 1: 1.5–1.8 s; Exp. 2: 1.5–1.9 s), after which a cue was centrally presented for 200 ms. For 16 participants (original Exp. 1), the cue consisted of a small circle with a black line pointing toward the upcoming target location. For the other 26 participants (original Exp. 2), the cue was an eight-legged blue cue, of which one leg turned red (or vice versa, counterbalanced, colors equated in luminance) indicating the upcoming target location. Apart from the visual properties of the cue and slight differences in intertrial intervals, the

procedures were nearly identical. After a delay period (600 ms), the search array was presented (200 ms), consisting of a single diamond target and seven distractor squares. Participants indicated whether the target diamond had a gap on the left or right side [20,37].

Participants completed 32 blocks, which consisted of 64 trials each. However, due to glitches and experimenter error, two participants completed 30 blocks, one participant completed 27 blocks, one participant completed 24 blocks, and one participant completed 36 blocks (of which 31 were recorded; see [35] for full details). The starting block cue condition (covert or neutral) was counterbalanced across participants.

### EEG acquisition and processing

**Experiment 1.** EEG was recorded using 20 tin electrodes in an elastic cap (Electro-Cap International, Eaton, OH). In addition to recording from International 10/20 sites F3, Fz, F4, T3, C3, Cz, C4, T4, P3, Pz, P4, T5, T6, O1, and O2, five electrodes were placed on non-standard sites. These sites were OL (midway between T5 and O1), OR (midway between T6 and O2), PO3 (midway between OL and P3), PO4 (midway between OR and P4), and POz (midway between OL and OR). The left mastoid was used as an online reference, and all data were rereferenced to the algebraic average of the left and right mastoids offline. Horizontal (1 cm from the outer canthi of the left and right eye) and vertical EOG (below the right eye) were recorded to monitor eye movements and blinks. EEG and EOG were recorded at 250 Hz with a 0.1-80 Hz bandpass filter using LabVIEW 6.1 (National Instruments, Austin, TX).

**Experiment 2.** EEG was recorded using 30 Ag/AgCl electrodes in an elastic cap (Brain Products). Data was recorded from International 10/20 sites Fp1, Fp2, F7, F3, Fz, F4, F8, FC5, FC1, FC2, FC6, C3, Cz, C4, CP5, CP1, CP2, CP6, P7, P3, Pz, P4, P8, PO7, PO3, PO4, PO8, O1, Oz, and O2. The right mastoid was used as an online reference, and all data were rereferenced to the algebraic average of the left and right mastoids offline. HEOG and VEOG electrodes were placed as in Experiment 1. EEG and EOG were recorded at 500 Hz with a 0.1–80 Hz bandpass filter using BrainVision Recorder (Brain Products).

**Artifact detection.** For artifact detection, we used identified artifacts from the original papers [34,35]. Artifact detection was based on visual inspection guided by an automated algorithm. Trials were checked for amplifier saturation, blinking, eye movements, excessive muscle noise or skin potentials. Moreover, we excluded the same noisy/bad channels from the analyses as reported in the original publications. We additionally excluded the first trial of each block, and trials where participants did not respond within 1950 ms of array onset (to exclude time-out trials). As the procedures for the first and second datasets from Foster, Bsales, and Awh [35] were identical apart from the employed cues, we collapsed these data to increase statistical power for our analyses.

**Preprocessing.** Data processing and analyses were performed using custom Python scripts and were based on previous work [34,37,43,52,59]. We here analyzed voltage patterns following previous decoding work [37,39,43,71,72]. We baseline corrected voltage activity by subtracting the mean voltage 250 ms prior to search array onset for each electrode per trial. We then applied a 6 Hz lowpass filter (as in [43]). Specifically, we used a zero-phase FIR filter, which applies the filter both forward and backward to eliminate phase shift, and consequently does not shift the data across time (using *filter_data* from the *mne* package). These baseline-corrected and filtered data served as the input to the decoding analyses.

### Inverted encoding models

To investigate whether previous target locations were neurally reactivated, we employed IEM [50,51]. We opted for IEMs as these have been effectively used to track spatial attention across space and time [34–36,54]. As we were interested in the reactivation of previous target locations, for IEM analyses only, we excluded trials where the previous target location matched the current target location (i.e., intertrial target distance of 0). Thus, our results cannot simply be attributed to the decoding of the current target location.

We reconstructed spatially sensitive channel tuning functions (CTFs) to the previous target location from the voltage topography. We assumed voltage activity at each electrode to reflect the weighted average of eight spatial channels that were each selective to a specific stimulus location. We modeled basis CTFs that reflected the above assumption using a half sinusoid raised to the 7th power which were each centered on one out of eight possible target locations (as in [34,37,54,73]).

For each iteration, we split the data into training (2 sets) and test (1 set) sets. Each set consisted of an equal number of trials per condition as well as an equal number of trials for each possible previous target location within a condition. Basis CTFs were fit to the training data using a general linear model (using *numpy.linalg.lstsq*) to obtain weights at each electrode. The obtained weights matrix was then inverted, which enabled us to reconstruct spatially-sensitive CTFs separately per condition. The procedure was conducted for each sample across time (every 4 or 2 ms, for Experiments 1 and 2, respectively), and to include all trials, we ran 100 iterations. To increase interpretability, we recentered CTFs (i.e., so the center corresponds to the previous target location) and averaged the reconstructed CTFs across the eight possible locations.

Following previous work [34,54], we quantified the spatial selectivity of the reconstructed CTFs by calculating their slope. More specifically, we used linear regression to estimate the slope of the reconstructed CTF (i.e., we averaged channel responses equidistant to the target location). A steeper CTF slope indicated a higher degree of spatial selectivity.

## Representational similarity analysis

The motivation for the RSA was twofold. First, we wanted to rule out possible influences of the current target location on previous target decoding more directly (also see S5 Fig). Second, RSA allowed us to simultaneously track isolated neural signals tracking the current and previous target.

RSA is built on the assumption that when conditions are more similar, neural activity (or another measure) should also be more similar [55]. For example, attending a target on the right should evoke similar activity to when one attends a target on the top right. By contrast, attending a target on the left would evoke highly dissimilar patterns in neural activity when compared with attending a target on the right [56–58]. Based on this, we constructed theoretical RDMs for the current and prior target locations (Fig 3) by calculating the distance (ranging from 0 to 4) between any two locations. We discarded previous current target combinations if these contained less than 4 trials. As previous and current targets were fully independent, the two theoretical RDMs were not correlated ($r = -.03$). Prior to calculating empirical distance, we binned voltage data in non-overlapping bins of 20 ms to increase the signal-to-noise ratio [52,59].

We used cross-validated Mahalanobis distance, a measure of multivariate distance, to quantify neural/empirical similarity across conditions [60]. We opted for Mahalanobis distances as this has been shown to be a reliable and unbiased measure of neural/empirical similarity [60], does not assume linearity, and is consistent with prior work [52,59]. Following Jones and colleagues [52,59], we computed cross-validated Mahalanobis distance by first splitting averaged voltage data for each electrode into train (50%) and test (50%) sets in a balanced manner so that the number of trials per previous–current location combination was the same. Next, each trial was demeaned by its previous–current location combination mean, and we subsequently calculated the variance–covariance matrix across the training trials (to prevent potential train-test leakage). The variance–covariance matrix was then regularized using the Ledoit–Wolf procedure, which finds the optimal shrinkage coefficient in a data-driven fashion [60,74]. To include all trials in our empirical dissimilarity matrices, we repeated this procedure for 1,000 train-test splits.

Equipped with both the theoretical (i.e., predictors) and empirical (i.e., outcome) RDMs, we were able to test whether previous and current target positions were represented in the neural data. As we did not assume any linear relationships, we used semi-partial rank correlations [52,59,61]. Semi-partial rank correlations isolate the relationship between a predictor and the outcome while controlling for the influence of another predictor. In our case, this allowed us to isolate the relationship between the location of the previous (current) target and the neural data while accounting for the influence of the

current (previous) target location. The obtained semi-partial rank correlations are a standardized measure of the strength of the influence of a predictor on an outcome variable, and therefore we were able to compare how strongly current and previous target locations were neurally represented. We repeated the full RSA procedure across all time bins.

## Statistical analysis

For the analysis of behavioral data, we used repeated-measures ANOVAs (using JASP v0.18.1). Greenhouse–Geisser correction was applied whenever the assumption of sphericity was violated. We calculated average accuracies and median response times per condition and split further based on the distance between the current and the previous target using artifact-free trials. To obtain an estimate of the spatial gradient of behavioral priming, we fit a linear regression for each condition separately per participant. These values, i.e., response time and accuracy slopes, reveal how spatially selective the intertrial priming effect is (akin to CTF slopes). We tested whether these slopes were reliably above zero using one-sample $t$-tests. We report $\eta_p^2$ and Cohen's $d_z$ as measures of effect size [75].

We tested whether CTF slopes were reliably above 0 (indicating reliable spatial selectivity) and whether semi-partial correlations were reliably above 0 (indicating reliable representation of a theoretical RDM) across participants. To circumvent the issue of multiple comparisons, we used cluster-based permutation tests [76]. Specifically, we used one-tailed tests (using *permutation_cluster_1samp_test* from the *mne* package) to identify cluster candidates and test their significance against 0 separately for each condition. To compare spatial selectivity across conditions, we calculated the difference in CTF slope between conditions and used a two-tailed one-sample cluster-corrected permutation test to assess significance. All cluster-based tests were conducted at $\alpha = .05$ using 10,000 iterations. We note that identified significant clusters are not a reliable measure of when an effect precisely occurs, but we chose to report the cluster sizes as recommended by Sassenhagen and Draschkow [77].

## Supporting information

**S1 Fig. Inverted encoding modeling (IEM) results with stickfunction basis channel responses.** Although we ensured that previous and current target locations never matched in the IEM analyses, we here conducted an additional control analysis to ensure current target locations did not influence our previous target location decoding results. To this end, we reconducted the IEM analyses using different basis functions wherein we did not assume a spatially-tuned profile but instead functions that predicted a response at the previous target location, and no response at all other locations. We found highly similar findings as those reported in the main paper for Experiment 1 (left) and 2 (right), ruling out the possibility that our previous target location decoding was driven by the location of the current target. Horizontal colored lines corresponding to conditions represent clusters where CTF slopes are reliably higher than 0 ($p < .05$). Horizontal black lines indicate significant differences between covert and neutral trials ($p < .05$).
(TIF)

**S2 Fig. Inverted encoding modeling results using the derivative of the voltage signal (i.e., change in voltage from sample to sample).** Importantly, this signal does not need any baseline correction, enabling us to examine if voltage changes after stimulus onset were sensitive to the previous target location. We found highly similar findings as those reported in the main paper for Experiment 1 (left) and 2 (right), ruling out the possibility that our previous target location decoding by the baseline correction procedure employed in the main paper. We also found that the reactivation in the neutral condition was stronger upon array onset than in the covert condition. This can be explained by considering the nature of the derivative data: based on the results reported in the main paper, it is likely that the prior target location remains activated within the cue-delay period in the covert condition, but if it is coded in a stable manner this would not be decodable using the derivative voltage signal. In other words, the previous target is likely already active in the EEG signal prior to array onset in the covert condition, and therefore the derivative response to the search array holds less information about

the prior target location than in the neutral condition. Horizontal colored lines corresponding to conditions represent clusters where CTF slopes are reliably higher than 0 ($p < .05$). Horizontal black lines indicate significant differences between covert and neutral trials ($p < .05$).
(TIF)

**S3 Fig. Eye-movement control analysis for Experiment 1.** To rule out eye movements as a possible confound for the decoding analysis of the previous target location reported in the main paper, we grouped trials in bins of two prior target locations left, right, up, or down. We first baseline corrected HEOG and VEOG data using the median voltage 250 ms prior to array onset. We then compared HEOG between trials where the prior target locations were on the left or on the right. We then compared VEOG between trials where the prior target locations were up- or downward. Neither of these analyses showed evidence that the prior target location was tracked by eye movements. HEOG (left) is plotted separated by trials where the prior target location was on the left or on the right. VEOG (right) is plotted separated by trials where the prior target location was up- or downward. Note that while HEOG was recorded using a bipolar montage (i.e., an electrode placed at the outer canthi of each eye), VEOG was recorded with a single electrode with a mastoid reference. This likely accounts for the different HEOG and VEOG responses across time. Nonetheless, comparisons between right and left, as well as up and down are still relevant to examine whether gaze tracked the previous target location. Error bands reflect standard errors of the mean.
(TIF)

**S4 Fig. Gaze analyses for the covert (left) and neutral (right) conditions of Experiment 2 ($n = 41$ out of 42 participants).** One participant was excluded due to unusable eye-tracking data. In the covert condition, gaze reliably pointed toward the cued location. Gaze seemed to point away from the previous target location, but this was not reliable across direction comparisons. In the neutral condition, gaze did not reliably point toward the current nor the previous target location. This rules out the possibility that gaze fully accounts for the observed neural decoding results. Horizontal black lines indicate significant differences between direction conditions ($p < .05$).
(TIF)

**S5 Fig. Gaze bias control analysis for Experiment 2 ($n = 41$ out of 42 participants).** We created two bins of trials wherein gaze was either biased toward or away from the previous target location. Specifically, for each participant, condition and location separately, we median-split trials based on how close gaze was from the previous target location (averaged 500–1,250 ms after cue onset). As can be seen in **A** and **B**, this created considerable gaze biases toward or away from the previous target location (pie plots correspond to previous target locations). Then, we reconducted the inverted encoding modeling procedure as in the main paper, but we now trained on the average neural data across the toward and away trials. This way, opposing gaze biases should be effectively canceled out and can no longer affect the neural decoding results. **C** Our results demonstrate that previous target locations were reactivated upon cue onset, and later upon array onset. Crucially, this occurred in both toward and away biased trials, and we found no differences between these conditions. This rules out miniature gaze biases as a confound to our neural decoding results of the previous target location. Horizontal orange and pink lines indicate significant CTF slopes above 0 ($p < .05$). All error bars and bands reflect standard errors of the mean.
(TIF)

**S6 Fig. Inverted encoding modeling results of the $n$-2 target location for the easy (blue) and hard (green) conditions.** As intertrial priming extends beyond the previous trial, we also tested whether the target location from two trials ago was reactivated. In this analysis, we excluded trials wherein the $n$-2 target location matched the current or the previous target location. In the easy search condition, we found two brief clusters that were significant ($p = .038$, $p = .019$). We found no significant reactivation of the $n$-2 target location in the hard search condition ($p$s > .18). There were no significant

differences between conditions. Thus, we found no strong evidence for a reactivation of the target position from two trials ago across search conditions. These results may suggest that these experiences are not (strongly) reactivated, or that EEG is not powerful enough to detect such reactivations across search conditions. Horizontal blues lines indicate significant spatial selectivity in the easy condition ($p < .05$). Error bands reflect standard errors of the mean.
(TIF)

**S1 Data. Intertrial Priming Affects Search in a Spatially Graded Fashion.** Response time (Fig 1B) and accuracy (Fig 1C) data ($n = 23$) across intertrial target distances separately for each individual from Experiment 1.
(XLSX)

**S2 Data. Intertrial Priming in A Cued Selection Task.** Response time (Fig 2B) and accuracy (Fig 2C) data ($n = 42$) across intertrial target distances separately for each individual from Experiment 2.
(XLSX)

## Acknowledgments

We thank Joshua J. Foster and colleagues for making the data from the original publications openly available. We thank Henry M. Jones for making the code for the RSA analyses from Jones and colleagues [52] openly available online.

## Author contributions

**Conceptualization:** Damian Koevoet.

**Formal analysis:** Damian Koevoet, Dirk Van Moorselaar.

**Funding acquisition:** Stefan Van der Stigchel.

**Investigation:** Damian Koevoet, Dirk Van Moorselaar, Edward Awh, Stefan Van der Stigchel.

**Methodology:** Edward Awh.

**Resources:** Damian Koevoet, Edward Awh.

**Software:** Damian Koevoet.

**Supervision:** Edward Awh, Stefan Van der Stigchel.

**Visualization:** Damian Koevoet.

**Writing – original draft:** Damian Koevoet.

**Writing – review & editing:** Damian Koevoet, Dirk Van Moorselaar, Edward Awh, Stefan Van der Stigchel.

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
