## [Editor Report · Decision Letter 0]

10 Sep 2025

Dear Dr Koevoet,

Thank you for submitting your manuscript entitled "Simultaneous Neural Representations of Past and Present Goals" for consideration as a Research Article by PLOS Biology.

Your manuscript has now been evaluated by the PLOS Biology editorial staff and I am writing to let you know that we would like to send your submission out for external peer review.

Once your full submission is complete, your paper will undergo a series of checks in preparation for peer review. After your manuscript has passed the checks it will be sent out for review. To provide the metadata for your submission, please Login to Editorial Manager (https://www.editorialmanager.com/pbiology) within two working days, i.e. by Sep 12 2025 11:59PM.

Kind regards,

Christian

Christian Schnell, PhD

Senior Editor

PLOS Biology

cschnell@plos.org

---

## [Decision Letter · Decision Letter 1]

1 Dec 2025

Dear Mr Koevoet,

Thank you for your patience while your manuscript "Simultaneous Neural Representations of Past and Present Goals" was peer-reviewed at PLOS Biology. It has now been evaluated by the PLOS Biology editors, an Academic Editor with relevant expertise, and by several independent reviewers.

In light of the reviews, which you will find at the end of this email, we would like to invite you to revise the work to thoroughly address the reviewers' reports.

As you will see below, the reviewers find the study very interesting, but they raise a number of concerns that we would like you to address in revision, for example by providing more methodological details, additional analyses of the eye movement data, repetition data and the reactivation readouts. This includes the idea of using ICA to remove the influence of eye movements and checking if there is actually a period before the trial where information about the previously relevant item seems to to disappear.

Given the extent of revision needed, we cannot make a decision about publication until we have seen the revised manuscript and your response to the reviewers' comments. Your revised manuscript is likely to be sent for further evaluation by all or a subset of the reviewers.

**IMPORTANT - SUBMITTING YOUR REVISION**

*Re-submission Checklist*

*Published Peer Review*

*PLOS Data Policy*

*Blot and Gel Data Policy*

Sincerely,

Christian

Christian Schnell, PhD

Senior Editor

PLOS Biology

cschnell@plos.org

REVIEWS:

Reviewer #1: The present work by Koevoet and colleagues reports reanalyzed EEG data from two earlier publications, now with a focus on how previous-trial target locations affect visual search, and how it interacts with current-trial target location. Using elegant IEM and RSA analyses, they observe that previous-trial target locations get consistently reactivated during the next stimulus event (the next search array, as well as a cuing display) and this signal coexists with the current-trial target information. Overall, I thought that the manuscript was interesting, well-done and that it will make a substantial contribution to the literature. Yet, I also had a couple of concerns and questions, which I hope the authors can address.

Major

1) I appreciated the analysis of the eye-movement data for experiment 1, which yielded clear results. Yet, there are a couple of other things that I think deserve attention in this regard: a) no such results are provided for experiment 2, which seems like an oversight; b) while reassuring concerning eye movements with regard to the previous target location, participants seem to quite clearly move their eyes in general (but see also last minor point), and likely towards the current target location (which is hard, if not impossible, to fully avoid). I think it would also be important to report results with regard to the eye-movements towards the current target location. c) related to the previous points, I was surprised that I did not see any information in the method section about how eye-related activity may have been attenuated, artifact rejection aside. Why not use ICA to remove as much eye-related activity as possible from all EEG channels, in order to avoid a direct influence in the multivariate results? If I am not overlooking something, I feel that it would be important to still do that.

2) One of the most frequent words in the present manuscript is "reactivated". I can in principle agree with its use, but I think it would then still be helpful to show that at some point the previous target location was deactivated. Would it be possible to show that preceding the n trial (target or cue, respectively for Exp1 and Exp2), the previous target location could no longer be decoded from the data? I believe that there were relatively long inter-trial intervals, which might be beneficial here. It may not be relevant, but some of the results struck me as showing very early effects of previous-target location (as early as 90 ms for the RSA of Exp1, but see also first minor point below), and I was wondering whether there was any way in which the baselining procedure in combination with the onset of a new stimulus may have temporarily suppressed an ongoing signal, rather than reactivating it. I am not necessarily assuming that this will be the case at all, but I would find it useful to show.

3) I would appreciate it if the authors could check whether the behavioral results were also influenced by full repetitions (same target location and same target stimulus/response), which could inflate the target-distance effects, in line with event-coding theories etc. The easiest way to quickly explore this would be to limit the analysis to only target-alternation trials (where the required response on trial n is opposite from that of trial n-1). I think this would be entirely sufficient to report in the response letter, but since it is really easy to explore, I would like to see this data, even if some of the references in the intro might already say something to this effect in principle.

4) This is more medium than major, but for the cue-based current-target location information, how can this be separated from simply information about the nature of the cue itself? It is also a spatially-selective stimulus, pointing in the relevant direction. Not a major concern, though, just a question for clarification.

5) Would it be possible to report information on which EEG channels contributed most to the current results, or in any other way add a spatial dimension to the results?

Minor

1) I was a bit surprised by the 6-Hz low-pass filter. This is probably fine, even if somewhat at odds with the highly-resolved testing logic, but should probably at least be acknowledged concerning timing accuracy (assuming a non-causal filter, e.g. onset latencies will shift forward, right?).

2) Readers that are not steeped in the technique (myself included) may think that the differences in semi-partial correlation between current and previous target location are large to an extent that feels incomparable; this may be worth contextualizing.

3) I am wondering whether the authors have any idea for why the HEOG in Fig. S1 is just a moderate drift, whereas the VEOG shows a large-amplitude, and highly temporally-structured response, with a very early negative peak (already around 100 ms, which seems very fast). Are EOGs expressed as bipolar montages, or with regard to the mastoid reference? From the looks of it, it seems like the HEOG might be from a bipolar montage, and the VEOG from a mastoid reference, but I am just speculating.

Reviewer #2: In this manuscript, the authors investigated the interplay of goal-driven and history-driven influences on visual search. Using EEG decoding and representational similarity analysis (RSA), the authors demonstrate across two tasks that prior target locations are neurally reactivated even when task-irrelevant, and that current and previous targets are concurrently represented during visual search. This was consistent with the behavioral results which demonstrated intertrial priming effects that incorporated localized attentional interference. There are a number of novel findings presented in the work and I think it makes an important contribution to attention research, particularly those who study visual search and selection history. I think the research was well designed, methodologically rigorous, and the conclusions follow logically from the findings. As such, I only have some queries for the authors and one minor recommendation for revision. To be clear, I don't feel that the revision is strictly necessary, in my view there are no significant issues with the work, but I have recommended minor revisions here so that the authors have an opportunity to incorporate this change if they choose to.

Minor revision:

In general I found the discussion to be quite cursory. There are some interesting ideas here, particularly the "operate-to-reactivate" section but they are explored only briefly and don't read as if they have been thoroughly considered.

In particular, one of the most interesting parts of the results to me was the fact that the reactivation of prior targets was lower in the neutral cue condition than the covert cue condition. The introduction of a goal-driven attentional influence in the covert condition was framed as something that, if anything, should decrease the reactivation of prior targets. The results seem to show the reverse. Relative to conditions with a neutral cue, covert spatial attention toward a new target location increased the reactivation of prior targets. This is seen numerically in the IEMs (and briefly statistically significantly following the array presentation in Figure 2D), but is also supported in the RSA analysis where the previous target location was not significantly reactivated in the neutral condition. While the conclusions seem to be suggesting that prior target representations are reactivated "despite covert spatial attention," these results could also be used to suggest that prior target reactivation in this task was enhanced by it. Aside from a sentence relating this to the "operate-to-reactivate" idea, this pattern of results isn't really discussed.

That is the main example of a discussion point that I felt went under-explored, but I do feel that this applies more generally int the discussion of the paper. Ideas are introduced very briefly and any evidence to support them is also brief. Some prior work is cited to support some of the ideas, but isn't really explained or discussed in a detailed way.

General queries:

1.I was interested to see that the authors applied a 6 Hz low-pass filter during preprocessing. I'm curious as to the rationale for this threshold as it seems much lower than anything I would usually expect to see. Is this something specific to IEMs?

2.In the IEMs, would adjacent target locations be more likely to be classified as one another? In these analyses you removed trials where target location had been repeated, to avoid having your "reactivation" driven by trials where the model simply decodes the current target location. However, if adjacent target locations are more similar than distant target locations, then doesn't this imply your effects could be driven by trials where the current target is adjacent to the previous target? The effect wouldn't be as strong as for two targets in the exact same location, but could this still produce an above chance effect? I understand that this doesn't explain all of your results, so your main conclusions likely stand, but might it explain why target representations disappear in the RSA analysis for the neural cue condition (where, to my understanding, this problem would be accounted for)?

Overall, I think this is a well-executed study that adds some innovative methods to the field and several novel findings that advance our understanding of attention. As mentioned, I found relatively little that I feel needs revising and would leave it to the authors discretion. The study appears to be methodologically rigorous, and of interest to researchers in cognitive neuroscience and related disciplines

---

## [Decision Letter · Decision Letter 2]

27 Mar 2026

Dear Dr Koevoet,

Thank you for your patience while we considered your revised manuscript "Simultaneous Neural Representations of Past and Present Goals" for publication as a Research Article at PLOS Biology. This revised version of your manuscript has been evaluated by the PLOS Biology editors, the Academic Editor and one of the original reviewers.

Based on the reviews and on our Academic Editor's assessment of your revision, we are likely to accept this manuscript for publication, provided you satisfactorily address the following data and other policy-related requests:

* We would like to suggest a different title to improve its accessibility for our broad audience:

"Past and present goals are represented concurrently during visual search"

* Please add the links to the funding agencies in the Financial Disclosure statement in the manuscript details.

* DATA POLICY:

Regardless of the method selected, please ensure that you provide the individual numerical values that underlie the summary data displayed in the following figure panels as they are essential for readers to assess your analysis and to reproduce it: 1BC and 2BC.

* CODE POLICY

Per journal policy, if you have generated any custom code during the course of this investigation, please make it available without restrictions. Please ensure that the code is sufficiently well documented and reusable, and that your Data Statement in the Editorial Manager submission system accurately describes where your code can be found. More information on our Code Policy, what and how to share can be found here: https://journals.plos.org/plosbiology/s/code-availability

We expect to receive your revised manuscript within two weeks.

*Published Peer Review History*

*Press*

Sincerely,

Christian

Christian Schnell, PhD

Senior Editor

cschnell@plos.org

PLOS Biology

Reviewer remarks:

Reviewer #1 (Nico Boehler signed his report): The authors have undertaken a very thorough revision, in particular providing many elegant control analyses; in my view, this has further strengthened an already compelling initial submission. I have no further comments.

---

## [Editor Report · Decision Letter 3]

8 Apr 2026

Dear Dr Koevoet,

Thank you for your patience while we considered your revised manuscript "Past and present goals are represented concurrently during visual search" for publication as a Research Article at PLOS Biology. This revised version of your manuscript has been evaluated by the PLOS Biology editors.

As discussed, I am returning the manuscript to you so you can upload the modified manuscript files with the correct labels.

As you address this point, please take this last chance to review your reference list to ensure that it is complete and correct. If you have cited papers that have been retracted, please include the rationale for doing so in the manuscript text, or remove these references and replace them with relevant current references. Any changes to the reference list should be mentioned in the cover letter that accompanies your revised manuscript.

We expect to receive your revised manuscript within two weeks.

*Published Peer Review History*

*Press*

Sincerely,

Christian

Christian Schnell, PhD

Senior Editor

cschnell@plos.org

PLOS Biology

---

## [Editor Report · Decision Letter 4]

10 Apr 2026

Dear Dr Koevoet,

Thank you for the submission of your revised Research Article "Past and present goals are represented concurrently during visual search" for publication in PLOS Biology. On behalf of my colleagues and the Academic Editor, Frank Tong, I am pleased to say that we can in principle accept your manuscript for publication, provided you address any remaining formatting and reporting issues. These will be detailed in an email you should receive within 2-3 business days from our colleagues in the journal operations team; no action is required from you until then. Please note that we will not be able to formally accept your manuscript and schedule it for publication until you have completed any requested changes.

PRESS

We frequently collaborate with press offices. If your institution or institutions have a press office, please notify them about your upcoming paper at this point, to enable them to help maximize its impact. If the press office is planning to promote your findings, we would be grateful if they could coordinate with biologypress@plos.org. If you have previously opted in to the early version process, we ask that you notify us immediately of any press plans so that we may opt out on your behalf.

Sincerely,

Christian

Christian Schnell, PhD

Senior Editor

PLOS Biology

cschnell@plos.org